# Interrelation of Hypoxia-Inducible Factor-1 Alpha (HIF-1 α) and the Ratio between the Mean Corpuscular Volume/Lymphocytes (MCVL) and the Cumulative Inflammatory Index (IIC) in Ulcerative Colitis

**DOI:** 10.3390/biomedicines11123137

**Published:** 2023-11-24

**Authors:** Ioan Sabin Poenariu, Lidia Boldeanu, Bogdan Silviu Ungureanu, Daniel Cosmin Caragea, Oana Mariana Cristea, Vlad Pădureanu, Isabela Siloși, Anca Marinela Ungureanu, Răzvan-Cristian Statie, Alina Elena Ciobanu, Dan Ionuț Gheonea, Eugen Osiac, Mihail Virgil Boldeanu

**Affiliations:** 1Doctoral School, University of Medicine and Pharmacy of Craiova, 200349 Craiova, Romania; sabinpoenariu@gmail.com (I.S.P.); statierazvan@gmail.com (R.-C.S.); elena.ciobanu210@gmail.com (A.E.C.); 2Department of Immunology, University of Medicine and Pharmacy of Craiova, 200349 Craiova, Romania; isabela_silosi@yahoo.com (I.S.); mihail.boldeanu@umfcv.ro (M.V.B.); 3Department of Microbiology, University of Medicine and Pharmacy of Craiova, 200349 Craiova, Romania; ioneteoana@yahoo.com (O.M.C.); ancaungureanu65@yahoo.com (A.M.U.); 4Department of Gastroenterology, University of Medicine and Pharmacy of Craiova, 200349 Craiova, Romania; bogdan.ungureanu@umfcv.ro (B.S.U.); digheonea@gmail.com (D.I.G.); 5Department of Nephrology, University of Medicine and Pharmacy of Craiova, 200349 Craiova, Romania; caragea.daniel87@yahoo.com; 6Department of Internal Medicine, Faculty of Medicine, University of Medicine and Pharmacy of Craiova, 200349 Craiova, Romania; 7Department of Biophysics, University of Medicine and Pharmacy of Craiova, 200349 Craiova, Romania; e_osiac@yahoo.com; 8Medico Science SRL—Stem Cell Bank Unit, 200690 Craiova, Romania

**Keywords:** ulcerative colitis, hypoxia-inducible factor-1α, IIC, cumulative inflammatory index, MCVL, average crepuscular volume−lymphocyte ratio, hematologic markers

## Abstract

We intended to investigate the presence and medical application of serum hypoxia-inducible factor-1 alpha (HIF-1α) along with the already known systemic inflammatory markers and the new one’s inflammatory indices, the proportion of mean corpuscular volume and lymphocytes (MCVL) and the cumulative inflammatory index (IIC), for patients with ulcerative colitis (UC). We sought to establish correlations that may be present between the serum levels of HIF-1α and these inflammatory indices, as well as their relationship with disease activity and the extent of UC, which can provide us with a more precise understanding of the evolution, prognosis, and future well-being of patients. Serum samples were collected from 46 patients diagnosed with UC and 23 controls. For our assessment of the serum levels of HIF-1α, we used the Enzyme-Linked Immunosorbent Assay (ELISA) technique. Thus, for HIF-1α we detected significantly higher values in more severe and more extensive UC. When it came to MCVL and IIC, we observed statistically significant differences between the three groups being compared (Severe, Moderate, and Mild). Our study highlighted that HIF-1α correlated much better with a disease activity score, MCVL, and IIC. With MCVL and IIC, a strong and very strong correlation had formed between them and well-known inflammation indices. By examining the ROC curves of the analyzed parameters, we recognized that TWI (accuracy of 83.70%) provides the best discrimination of patients with early forms of UC, followed by HIF-1α (73.90% accuracy), MCVL (70.90% accuracy), and PLR (70.40%). In our study, we observed that HIF-1α, MCVL, and PLR had the same sensitivity (73.33%) but HIF-1α had a much better specificity (60.87% vs. 58.70%, and 54.35%). Also, in addition to the PLR, HIF-1α and MCVL can be used as independent predictor factors in the discrimination of patients with early forms of UC.

## 1. Introduction

Ulcerative colitis (UC) is generally regarded as a chronic, inflammatory, idiopathic condition. Although the primary effects are seen in the gastrointestinal tract, it can likewise present with important extraintestinal manifestations, at the skin, joint, liver, and eye level.

UC manifests itself in people with genetic susceptibility, involving environ-mental factors that interact with the intestinal microflora and generate an inadequate individual intestinal immune response [1]. Both innate and adaptive immune response defects lead to intestinal inflammation and epithelial injury. Significant evidence has been reported highlighting that these inflamed lesions of the intestinal mucosa are intensely hypoxic [2,3,4]. It is also considered that this inflammatory hypoxia (the term used for tissue inflammation leading to tissue hypoxia) acts as a critical endogenous anti-inflammatory pathway that protects tissues [5,6,7,8,9].

Intestinal epithelial cells respond to hypoxia-induced low oxygen through oxygen-sensitive transcription factors such as hypoxia-inducible factor (HIF). This specific factor is constituted by an oxygen-sensitive substance, the α subunit (HIF-1α, HIF-2α, and HIF-3α), and the conserved β subunit (HIF-1β) [10]. There is a wealth of data that highlights the crucial role of HIF-1α in the pathogenesis of UC, as well as findings that indicate that HIF-1α can be utilized as a biomarker to assess the severity or overall disease activity in patients suffering from UC [11,12,13,14,15,16,17,18,19,20,21,22].

Noteworthy are the following hematological markers of systemic inflammation derived from the complete blood count panel (CBC): neutrophil-to-lymphocyte ratio (NLR), which highlights leukocytosis and lymphopenia in its initial inflammatory stage; monocyte-to-lymphocyte ratio (MLR), which contributes to the body’s immune status, with a decrease in its value indicating an immune dysfunction of the host; platelet-to-lymphocyte ratio (PLR), which is an inflammatory marker for diseases which are immune-mediated, metabolic, and pro-thrombotic in nature; the derived neutrophil-to-lymphocyte ratio (dNLR) (neutrophils/(white blood cells-neutrophils)), and the aggregate index of systemic inflammation (AISI) ((neutrophils × monocytes × platelets)/lymphocytes), which have been studied in Coronavirus disease-19 (COVID-19) [23,24]; the systemic immune-inflammation index (SII) ((neutrophils × platelets)/lymphocytes) [25,26,27,28]; and the systemic inflammatory response index (SIRI) ((neutrophils × monocytes)/lymphocytes) [29]. These hematological indices are considered to be faithful biomarkers for both sides of the inflammatory response, the innate immune system and the adaptive immune response, with increased sensitivity, accuracy, and availability, in addition to the traditional serological indicators (erythroyte sedimentation rate (ESR), fibrinogen (FIB), C-reactive protein (CRP) and white blood cell (WBC) count) [30,31,32,33,34,35,36,37]. AISI, SIRI, and SII are indicative of the immune and inflammatory status, which is why a link between them and the risk of mortality in various types of cancer, stroke, autoimmune diseases, as well as infectious diseases [24,38,39,40,41], has been observed.

Previous research endeavors have concentrated on the effectiveness of these non-invasive biomarkers for assessing the disease activity of UC [42,43,44,45,46,47] and Crohn’s disease [48,49], correlating with clinical or endoscopic disease activity indices, as well as predicting mucosal healing, the therapeutic outcomes, or clinical relapses.

Radulescu et al. [50] were the first to have highlighted new hematological indices calculated according to the increased values of the red cell distribution width (RDW), and the mean corpuscular volume (MCV), and they obtained the ratio between the mean corpuscular volume and lymphocytes (MCVL), and the cumulative inflammatory index (IIC) ((mean corpuscular volume × width of erythrocyte distribution × neutrophils)/(lymphocytes × 1000)). These new hematological indices better reflect the inflammation-induced changes to red and white cells in the peripheral blood following an inflammatory process. The authors started from the observation that the deceased patients had a significantly higher RDW value, both in the pre-Coronavirus disease (pre-COVID) and in the peri-Coronavirus disease (peri-COVID) group, and that the MCV had values higher than 100 in the patients who died in the peri-COVID group. Using these markers that summarize red blood cell (RBC) changes that can arise in either acute or chronic inflammation, they associated them with numerical changes in neutrophils, which are components of the innate immune system, and lymphocytes, which serve as markers for the adaptive immune response, thus establishing a new prognostic marker of the severity of acute pancreatitis patients, namely IIC [50].

The literature so far has referred to the diagnostic value of some of these hematological markers, NLR, MLR, and PLR, in UC. However, we could not find any studies reporting on the correlation between these biomarkers and the level of hypoxia. Therefore, we focused on the evaluation of hypoxia by determining the presence and clinical use of serum HIF-1α in a cohort of patients with UC and their relationship with the disease activity and the extent of UC (Montreal classification). We also wanted to identify the possible correlations between the serum levels of HIF-1α with disease activity of UC, and the status of systemic immune inflammation using well-known markers (NLR, MLR, PLR, dNLR, AISI, SII, SIRI, hs-CRP, ESR) and new ones, specifically MCVL and IIC, which can provide us with more insight into the evolution, outlook, and forthcoming quality of life for such patients. The study’s third relevant objective was to assess the diagnostic accuracy of HIF-1α, NLR, MLR, PLR, dNLR, AISI, SII, SIRI, as well as the recently introduced MCVL and IIC as separate predictive factors, using receiver-operator characteristic (ROC) curve analysis.

## 2. Materials and Methods

### 2.1. Study Design

This study was carried out following approval from the Committee of Ethics and Academic and Scientific Deontology of the University of Medicine and Pharmacy from Craiova, number 146/8 July 2022. Every patient signed an informed consent form to participate in our study. All the procedures were followed in compliance with the ethical standards of the institutional responsible committees for human studies, the Helsinki Declaration of 1975–2013, as revised in 2008.

### 2.2. Patient Selection

This prospective study was conducted in the Gastroenterology Clinic, Emergency County Clinical Hospital of Craiova.

We included in the study eighty-nine patients diagnosed with UC (UC group) aged between 27 and 71 years. At visit 1, the patients were clinically examined and explored through lower digestive endoscopy and histopathological examination of the biopsy samples. Patients with UC were followed for 6 months. Of the eighty-nine patients diagnosed with UC, forty-six completed the study and were included in the final evaluation, while forty-three patients were lost to follow-up due to associated diabetes mellitus (*n* = 5), clinically evident infections (*n* = 6), liver disease (*n* = 4), unwillingness to continue (*n* = 5), relocation (*n* = 4), patients that achieved remission (*n* = 10), and patients that were receiving tumor necrosis factor-alpha (TNF-α) inhibitors either as monotherapy or in combination with immunomodulators (*n* = 9) (Figure 1). At visit 2 (after 6 months), the patients were clinically re-evaluated.

For the comparative analysis, a control group (patients without active UC or in the antecedent stages) was formed comprising 23 healthy subjects (C group) selected from the patients who had undergone a routine health checkup at the Gastroenterology Clinic and had a normal colonoscopy exam.

Groups were formed based on specific criteria. The criteria for inclusion in the study were patients diagnosed with active UC and aged over 18 years. For the control group, the inclusion criteria were: patients without active or previous UC, and a compatible group in terms of age and male/female ratio with the UC group. Patients with infection, neoplastic disorders, another organic disease (heart disease, pulmonary disease, hepatosplenic disease, renal insufficiency), hematologic disease, metabolic disease, or other autoimmune diseases were excluded from the study.

For each patient with UC, a structured form was drawn up that included contact information, demographic data, hereditary and personal pathological antecedents, clinical manifestations, laboratory analyses, phenotypic classification, intestinal and extraintestinal complications, disease duration, operation history, classes of drugs used, and the scheme of current treatment.

The UC group received 5-aminosalicylates and immunomodulatory therapies, including thiopurine, methotrexate, cyclosporine, and tacrolimus.

The patients with UC were included in the Montreal classification [51], according to the endoscopic extent of the disease: E1—proctitis, E2—left colitis, E3—extensive colitis. As a method for assessing disease activity, we used the Truelove and Witts severity index (TWI). Patients were categorized into the appropriate activity type based on the score they achieved, thus: TWI score ≤ 5, corresponding to a Mild (Mi) disease activity (*Bowel movements (number per day)*—Fewer than 4; *Blood in stools*—No more than small amounts of blood; *Pyrexia (temperature higher than* 37.8 °C)—No; *Pulse rate higher than* 90 bpm—No; *Anaemia*—No; *ESR* (mm/hour)—30 or below); TWI score = 5–9, corresponding to a Moderate (Mo) disease activity (*Bowel movements (number per day)*—4–6; *Blood in stools*—Between mild and severe; *Pyrexia (temperature greater than* 37.8 °C)—No; *Pulse rate greater than* 90 bpm—No; *Anaemia*—No; *ESR* (mm/hour)—30 or below); TWI score ≥ 9, corresponding to a Severe (S) disease activity (*Bowel movements (number per day)*—6 or more along with a minimum of 1 systemic upset aspect (marked with * below): *Blood in stools*—Visible blood; *Pyrexia (temperature greater than* 37.8 °C)—Yes; *Pulse rate greater than* 90 bpm—Yes; *Anaemia*—Yes; *ESR* (mm/hour)—Above 30) [52,53].

### 2.3. Sample Collection

The samples were collected at visit 2 in both groups (UC and C). The biological samples gathered from patients consisted of venous blood (approximately 5 mL) collected in additive-free tubes (Becton Dickinson vacutainer, Franklin Lakes, NJ, USA). After harvesting, the clot was separated through centrifugation (Hermle AG, Gosheim, Baden-Württemberg, GE), 3000× *g* for 10 min, within 4 h of collection, following the standard protocol. The serum sample tubes were labeled for each patient, securely sealed to prevent contamination, and stored at temperatures ranging from −20 °C to −80 °C, enabling extended processing periods for the samples. Before processing patient samples, frozen specimens were allowed to thaw at room temperature, and freezing-thawing cycles were avoided.

Peripheral venous blood was drawn into vacutainer tubes containing ethylene-diamine-tetra-acetic acid (EDTA) anticoagulant and was utilized for conducting a CBC: white blood cells (neutrophil, lymphocyte, monocyte), red blood cells, platelet count, hemoglobin, and hematocrit.

### 2.4. Immunological Assessment

For the quantitative dosing of serum concentrations of the HIF-1 α, the Enzyme-Linked Immunosorbent Assay (ELISA) technique was employed at the University of Medicine and Pharmacy of Craiova’s Immunology Laboratory.

We utilized commercially available test sets tailored for each of the following mediators: HIF-1 α (Catalog# EHIF1A; sensitivity or Lower Limit of Detection: <30 pg/mL; assay range: 81.92–20,000.00 pg/mL), Invitrogen, Thermo Fisher Scientific, Inc. (Waltham, MA, USA).

After thawing, each sample underwent dilution following the manufacturer’s instructions and their prescribed method, with a common optical analyzer operating at a wavelength of 450 nm being utilized for the process.

### 2.5. Complete Blood Counting (CBC)

To differentiate and count all five types of white blood cells (neutrophils, monocytes, lymphocytes, basophils, and platelets), RDW, and MCV, an automatic hematology analyzer was utilized, giving us an extended leukocyte formula 5 diff, by way of flow cytometry and Coulter’s principle (Ruby Cell-Dyne, Abbott, Abbott Park, IL, USA). Through these determinations, the inflammation indices obtained from the blood cell count, NLR, MLR, PLR, dNLR, AISI, SII, SIRI, MCVL, and IIC, were calculated.

ESR was conducted by the Westergren method (ESR tubes, Becton Dickinson, Franklin Lakes, NJ, USA). The high-sensitivity CRP (hs-CRP) was calculated using an automatic immunoassay analyzer by the chemiluminescence immune method (Cobas e411, Roche Diagnostics GmbH, Mannheim, Germany).

### 2.6. Statistical Analysis

Patient data management and data processing were carried out utilizing Microsoft Excel and its Data Analysis module, while statistical analysis was conducted using the trial version of GraphPad Prism 5 (LLC, San Diego, CA, USA).

We assessed data normality using the D’Agostino and Pearson omnibus normality test. For variables that followed a normal distribution, we presented the mean value accompanied by its standard deviation (stdev), whereas categorical values were represented as percentages.

One-way ANOVA was used to analyze the differences between the groups for parametric variables, while the Kruskal—Wallis test was used for non-parametric variables. The existence of significant correlations between the concentration of the HIF-1 α, TWI, and different ratios of immunocytes (NLR, MLR, PLR, dNLR, AISI, SII, SIRI, MCVL, and IIC), were evaluated utilizing Pearson’s coefficients (−1 < r < 1), and visually represented through a correlation heatmap matrix. Strong positive correlations were depicted in bright blue, while strong negative correlations were shown in bright red. A *p*-value < 0.05 was considered statistically significant.

We assessed the diagnostic efficacy of the studied markers through the analysis of ROC curves. We quantified the performance using the area under the ROC curve (AUC) and p statistics, comparing the calculated AUC with the threshold of AUC = 0.5 (indicating a weak discriminative marker). Cut-off values, associated with the highest accuracy, were identified. We explored various threshold values for each marker, calculating both sensitivity and specificity.

## 3. Results

### 3.1. Clinical Characteristics of the Study Subjects

Out of the forty-six patients diagnosed with UC, 54.35% were male (sex ratio: 25 male/21 female), with a mean age of 49.30 years (stdev. 18.00). In the C group, the mean age was 54.30 years (stdev. 5.75). There was no notable disparity in age between the two groups (UC vs. C) (*p* = 0.158). Hence, both groups of subjects exhibited similarities, displaying homogeneous demographic data in terms of age, sex ratio during blood sampling, and residential area (rural or urban) (Table 1).

Depending on the residence criteria (rural or urban), we noticed that two-thirds (65.22%) of the patients diagnosed with UC came from urban areas. This phenomenon could be attributed to the rapid embrace of a Westernized lifestyle by urban patients, coupled with their easier access to healthcare services. The results obtained in our study are in accordance with those reported in other studies, conducted in both highly industrialized nations and regions where the standard of living is moving toward Westernization.

Regarding the location of the lesions, according to the endoscopic extent of the disease, left colon damage predominated (left-sided UC, Montreal classification E2) in a percentage of 39.13 (18 patients). Ulcerative proctitis (Montreal classification E1) was present in 15 patients (32.61%) and 13 patients (28.26%) had Extensive UC (pancolitis) (Montreal classification E3).

According to the TWI score for assessing disease activity, we observed that most patients presented moderate (22 patients, 47.83%) and mild (15 patients, 32.61%) activity, while in nine patients (19.57%) we identified a severe activity. Patients were categorized into three subgroups: Mild (Mi), indicated by a TWI score reflecting mild disease activity; Moderate (Mo), indicated by a TWI score showing moderate disease activity; and Severe (S), indicated by a TWI score demonstrating severe disease activity.

### 3.2. Concentrations of Serum HIF-1α and Disease Activity

The HIF-1α value (Table 2) exhibited a statistically significant increase in all test groups (Mi, Mo, and S) compared to the C group, with a *p*-value of less than 0.05.

Comparing the mean serum levels of HIF-1α, we obtained the following data: Mi vs. C group, 91.49 ± 42.21 vs. 51.90 ± 24.34 ng/L, *p* = 0.0057; Mo vs. C group, 157.20 ± 74.57 vs. 51.90 ± 24.34 ng/L, *p* < 0.0001; S vs. C group, 396.70 ± 86.51 vs. 51.90 ± 24.34 ng/L, *p* < 0.0001.

Serum levels of HIF-1α showed increases directly related to UC activity, with higher values among the S group. There were statistically significant differences between the serum levels of HIF-1α among the S vs. Mo group (*p* < 0.0001), S vs. Mi group (*p* < 0.0001), and Mo vs. Mi group (*p* = 0.0365) (Figure 2A).

### 3.3. Concentrations of Serum HIF-1α and Extent of Disease

Analyzing the group of UC patients according to the extent of UC (Montreal classification), we found that HIF-1α levels (Table 2) were higher in all the test groups (E1, E2, and E3) than in the C group with a statistically significant difference.

There were statistically significant differences between the serum levels of HIF-1α among patients with E2 vs. E1 extent (*p* = 0.0280), E3 vs. E1 extent (*p* = 0.0042), and E3 vs. E2 extent (*p* = 0.0082). In our study, we found that increased serum levels of HIF-1α were associated with the extent of intestinal lesions, lesions that lead to the intensification of hypoxia (Figure 2B).

### 3.4. The Inflammatory Status Biomarkers

Analyzing the levels of inflammatory status biomarkers in relation to the UC activity, we found statistically significant differences between the three groups (S, Mo, and Mi), as detailed in Table 3. The serum levels displayed a trend proportional to the changes in the disease activity, the most modified means of the levels being observed in the S group.

Regarding the new hematological indices we studied, our findings were as follows: for MCVL we observed statistically significant differences between the serum levels among S vs. Mo, and S vs. Mi group (1.67- and 2.26-fold, respectively) and Mo vs. Mi group (1.39-fold); while for IIC we obtained significantly higher levels between S vs. Mo and S vs. Mi group (1.17- and 1.29-fold, respectively).

Comparing the index values of the already known indices, we obtained the following statistically significant differences: NLR, between the serum levels among S vs. Mi group (1.40-fold); MLR, between the serum levels among S vs. Mi group (1.62-fold); PLR, between the serum levels among S vs. Mo and S vs. Mi group (2.30- and 4.16-fold, respectively), and Mo vs. Mi group (1.80-fold); AISI, between the serum levels among S vs. Mo and S vs. Mi group (1.48- and 1.67-fold, respectively); SII, between the serum levels among S vs. Mo and S vs. Mi group (1.71- and 2.13-fold, respectively); SIRI, between the serum levels among S vs. Mi group (1.32-fold).

Several studies have indicated that assessing the neutrophil-to-lymphocyte ratio (NLR) and the platelet-to-lymphocyte ratio (PLR) values can aid in the diagnosis of UC and the evaluation of its activity [30,33,34,35,36,37,54]. NLR and PLR values can predict UC activation and endoscopic mucosal lesions [27,33,54]. Thus, NLR and PLR values exhibited positive correlations with endoscopic scores. As such, Akpinar et al. [37] illustrated that NLR and PLR could discern and forecast active disease under endoscopy as outlined by the Rachmilewitz endoscopic activity index. Moreover, as demonstrated in the study conducted by Celikbilek et al. [35], NLR was identified to be associated with the onset of ulcerative colitis. Current research indicates a substantial increase in NLR levels among patients experiencing active ulcerative colitis [33,54,55]. Several studies have demonstrated that the NLR values in patients with UC were notably higher than those in the healthy group. Furthermore, the NLR values in patients with active UC were significantly elevated compared to those in patients with inactive UC [56,57,58]. Nevertheless, some studies have different opinions. In a retrospective case-control study conducted by Cherfane, it was noted that NLR values can accurately differentiate active UC from controls, but are not as effective in distinguishing inactive UC [36]. In a meta-analysis, Ma et al. [31] demonstrated that NLR serves as a dependable inflammatory marker, accurately reflecting disease status and assessing UC activity.

Analyzing the levels of inflammatory status biomarkers in relation to the extent of disease, we found statistically significant differences between the three groups compared (E1, E2 and E3), as detailed in Table 4. The serum levels had a trend proportional to the extent of disease, the most modified means of the levels showing up in the E3 group. Comparing the index values of the new hematological indices investigated by us, we found only MCVL showed statistically significant differences between the E3 vs. E1 group (1.30-fold).

Related to the already known indices, we obtained statistically significant differences between index values among the E3 vs. E1 group: NLR (1.04-fold), MLR (1.23-fold), SII (1.17-fold), and SIRI (1.20-fold). Statistically significant differences were observed between the index values of PLR among the E3 vs. E1 group (1.42-fold), and the E3 vs. E2 group (1.41-fold).

In our study, we also obtained an increased number of monocytes, directly proportional to the activity of the disease, with high values being in the S group, but also relating to the location of the lesions, with the pancolitis group having a higher number of monocytes. In a study published in 2015, Cherfane et al. [36] noted that increased absolute monocytic count and low lymphocyte-to-monocyte ratio (LMR) can forecast disease activity in UC patients. These findings are consistent with a study by Okba et al. [33] who identified a notable rise in absolute monocytic count and a reduction in LMR among active UC patients compared to those with inactive UC and controls. The increased presence of monocytes is expected in severe forms of the disease and during acute inflammation, as Shi et al. [59] showed in a study. This hypothesis was also suggested in a study by MacDonald et al. [60] which showed that persistent monocyte activation and impaired innate immune responses are mechanisms implicated in the onset of inflammatory bowel diseases.

In our study, we did not find statistically significant differences in the values of dNLR and AISI among the study groups, regarding both disease activity (S, Mo, and Mi) and the extent of the lesions (E1, E2 and E3), in patients with UC. The dNLR echoes the increase in the number of neutrophils and the relative decrease in the number of lymphocytes, hence indicating that inflammatory responses rely on neutrophils. This was first proposed by Takada et al. [61] in evaluating the prognosis of patients with metastatic disease regardless of the treatment followed. A second study by Radulescu et al. [50], comparing complications with surgical risk in the pre-COVID and peri-COVID periods, found that dNLR could predict the occurrence of complications and mortality.

Our study showed higher SII values in the S UC group compared to the Mo and Mi groups, and higher SII values in the E3 group compared to the E2 and E1 groups, with statistically significant differences. Similarly to the other studies, we demonstrated that the SII values were significantly correlated with the activity and extent of UC. The SII was proposed for the first time by Hu et al. in 2014, as being derived from neutrophils, platelets, and lymphocytes [39]. Increasing values for SII indicate a high number of neutrophils and platelets, so a higher inflammatory response from the body is associated with a low number of lymphocytes, which suggests a defective cellular immune response.

The first data related to the determination of SII in patients with UC were provided by Zhang et al. [25] in a study published in 2021, who determined for the first time SII in patients with UC and evaluated the diagnostic value of using SII in assessing the severity of the disease, compared to NLR or PLR. The authors observed elevated SII, NLR, and PLR values in UC patients compared to controls. These values exhibited positive correlations with the Mayo endoscopic score, extent of disease, Degree of Ulcerative Colitis Burden of Luminal Inflammation (DUBLIN) score, and Ulcerative Colitis Endoscopic Index of Severity (UCEIS). The study concluded that the SII could be used as a summary of information on the activity and degree of damage to the mucosa before the endoscopic examination, especially in patients with an elevated response to invasive investigations or in the case of inability to obtain equipment in the active time period. Xie et al. [26] revealed elevated SII values in the moderate and severe UC subgroups in contrast to the mild or remission subgroups. Furthermore, in this study, a correlation of SII values with Mayo score, CRP, and ESR was obtained. Pakoz’s study indicated that the SII value is elevated during active disease, attributed to the increase in platelet and neutrophil counts, aligning with the pathogenesis of ulcerative colitis [27].

Our research revealed statistically significant differences regarding higher SIRI values in the S UC group compared to the Mi group, and higher SII values in the E3 group compared to the E1 group. The SIRI was formulated by YU et al. [29], using the peripheral venous blood neutrophil, monocyte, and lymphocyte counts. The authors indicated that higher SIRI levels were significantly correlated with active inflammatory bowel disease (IBD), CRP, ESR, NLR, and MLR, and might be a novel promising marker of the disease severity in IBD.

### 3.5. Correlations between Serum Concentrations of HIF-1α, TWI, and Inflammatory Status Biomarkers

We analyzed the possible correlations between HIF-1α, TWI, and inflammatory status biomarkers, in relation to the extent of disease (Figure 3, Figure 4 and Figure 5).

In the E1 location of the lesions, HIF-1α values were moderately correlated to the limit of significance with MCVL (r = 0.514, *p* = 0.050). We established a notable positive correlation between the TWI score and both hs-CRP (r = 0.730, *p* = 0.002) and ESR (r = 0.780, *p* = 0.001). Strong and very strong positive significant correlations were observed between hematological markers, as shown in Figure 3.

In left colitis of the lesions (E2 location), HIF-1α correlated strongly and significantly with TWI score (r =0.710, *p* = 0.007) and with ESR (r =0.620, *p* = 0.0024). Furthermore, we observed a moderate yet statistically significant positive correlation between IIC and ESR (r = 0.576, *p* = 0.039) as well as a strong and statistically significant positive correlation between TWI score and ESR (r = 0.785, *p* = 0.001) (Figure 4).

We can observe that the new hematological indices investigated by us correlated strongly with each other (MCVL with IIC, r = 0.615, *p* = 0.025), as well as the fact that MCVL and IIC correlated strongly and very strongly with the already known inflammation indices (NLR, dNLR, SII, SIRI).

Analyzing the target parameters determined in the E3 group (pancolitis), we detected that HIF-1α correlated much better with TWI and inflammatory status biomarkers. Strongly significant positive correlation with HIF-1α was found with the TWI score (r = 0.767, *p* = 0.0001), hs-CRP (r = 0.804, *p* = 0.0001), MCVL (r = 0.663, *p* = 0.046), IIC (r = 0.585, *p* = 0.019), SIRI (r = 0.453, *p* = 0.036), and AISI (r = 0.455, *p* = 0.048), as shown in Figure 5.

Regarding the correlation of the serum level of HIF-1α with the severity of the disease evaluated by various indices, or endoscopic grades, the studies provided us with important information. A positive correlation was identified between the expression levels of HIF-1α in both serum and colonic mucosa and the severity of the disease course as well as the endoscopic grading in patients with UC, both in remission and during the acute phase of the disease [16]. The Pearson correlation analysis revealed a significant association between HIF-1α and CRP levels with the Rachmilewitz score and disease activity index (DAI) [17], and the level of serum HIF-1α was correlated with the activity of UC, mucosal healing, degree of inflammation, and oxidative stress [62], suggesting that the identification of HIF-1α expression could serve as a predictor for disease severity.

Furthermore, we observed that the new hematological indices we investigated correlated moderately with each other (MCVL with IIC, r = 0.467, *p* = 0.025), as well as the fact that MCVL and IIC correlated strongly and very strongly with the already known inflammation indices (NLR, dNLR, SII, SIRI, AISI).

In our research, SII values correlated strongly and very strongly with NLR, PLR, dNLR, and AISI in the E2 and E1 groups.

### 3.6. Diagnostic Accuracy of the Biomarkers

Our study aimed to evaluate, following the analysis of the ROC curve, whether the investigated parameters are able to discriminate between patients in the early forms of UC and those with moderate and severe forms. An optimal value established by maximizing the sum of sensitivity and specificity for each parameter was also determined.

Table 5 and Figure 6A–D display the ROC curves for the analyzed parameters. It is clear from these curves that TWI (accuracy of 83.70%) provides the best discrimination of patients with early forms of UC, followed by HIF-1α (73.90% accuracy), MCVL (70.90% accuracy), and PLR (70.40%).

The prediction of patients with early forms of UC using the ROC curve showed the following significant values: the cut-off value of the HIF-1α level was determined to be >119.80 with 73.33% sensitivity and 60.87% specificity (AUC = 0.739, *p* = 0.036); MCVL was determined to be >45.63 with 73.33% sensitivity and 58.70% specificity (AUC = 0.709, *p* = 0.025); PLR was found to be >129.40 with 73.33% sensitivity and 54.35% specificity (AUC = 0.704, *p* = 0.047); IIC, SIRI, SII, MLR, NLR, AISI, and dNLR had the area <0.700.

In our study, we observed that HIF-1α, MCVL, and PLR had the same sensitivity (73.33%) but HIF-1α had a much better specificity (60.87% vs. 58.70% and 54.35%) in discrimination of patients with early forms of UC.

## 4. Discussion

UC is considered a multifactorial disease. The precise causes and the complex underlying mechanisms that result in tissue destruction remain highly intricate and not fully understood up to this point.

Over the past few decades, scientific research on this disease has intensified, leading to the formulation of various theories and hypotheses to explain the underlying mechanisms. Significant progress has been made, allowing us to consider the following hypothesis as valid: UC occurs as a result of an insufficient immune response that develops in individuals with genetic susceptibility, and this response involves a complex interplay between evolving environmental changes driven by societal progress, the continually changing intestinal microflora, and the hyperactivation of the individual’s intestinal immune system [1,63]. This ends up disrupting homeostasis of the intestinal mucosa along with its barrier function, leading to immune-mediated tissue damage and the manifestation of clinical symptoms.

Experimental data and clinical observations, derived from studying disrupted immune mechanisms and intestinal inflammation in animal models, where a disease model was induced and developed, have enhanced our understanding of UC pathogenesis. These disrupted immune mechanisms stem from congenital or acquired immune system defects, triggered by changes in the intestinal microflora.

UC is characterized by repeated mucosal injury, disruption of the tight junction integrity of the intestinal epithelial barrier, which leads to increased intestinal permeability [22,64,65], inflammation, and ultimately bacteremia (elevated levels of serum endotoxin) [66].

Keely et al. [22] showed that HIF-1α directly oversees the activation of multiple protective genes in reaction to harm done to the integrity of the epithelial barrier, and induces a decrease in the production of cytokines as well as an increase in the production of β-defensins, with a key role in antimicrobial immunity.

According to the specialized literature, most of the studies that investigated the expression of HIF-1α and HIF-2α, along with the effect of the prolyl hydroxylase inhibitors of HIFs, were carried out on a large scale in experimental models, the so-called “chemical” models of UC. Oral or rectal administration of specific chemical agents, for example, Dextran Sodium Sulfate, 2,4,6-trinitrobenzene sulfonic acid, was followed by inflammation of the intestine that is similar in clinical manifestations and morphology to UC. Thus, these studies intended to explore the role of HIFs at the onset of UC, as well as their impact on the intensity of the inflammatory process [11,12,13,14,15,18,19,20,21,22].

After performing a database analysis, we were able to locate only three studies that had the following goals: determining the serum HIF-1α expression levels, looking into its regulatory role in pathogenesis, and evaluating its association with disease activity and severity in UC patients.

The initial study published in 2016 by Xu et al. [16] aimed to quantify HIF-1α in both serum and colonic mucosa of UC patients, analyzing its role in the pathogenesis, disease activity, and severity of UC. The serum HIF-1α level in UC patients was markedly elevated compared to both UC patients in remission and the control group. Furthermore, UC patients exhibited significantly elevated HIF-1α expression in colonic mucosa compared to both UC patients in remission and the control group. They observed an escalation in HIF-1α expression as the disease advanced.

The second study, conducted in another region in China [17] on a slightly larger group of patients, revealed that expression levels of HIF-1α and CRP were significantly higher in the UC group compared to the control group. Moreover, HIF-1α and CRP levels in UC patients increased significantly with the severity of the disease. Patients in the mild group exhibited the lowest levels of HIF-1α and CRP, while those in the severe group showed the highest levels of both markers.

In the most recent study published in 2022, Li S. et al. [62] obtained the same results. The serum HIF-1α level in active UC patients was markedly elevated compared to both the remission group and the control group, and the expression of HIF-1α varied significantly across different clinical severity and endoscopic manifestation grades.

We observed elevated HIF-1α levels in all test groups (Mi, Mo, and S) compared to the control group, with a statistically significant difference. Serum concentrations of HIF-1α showed to be increased directly related to UC activity, with higher values in the S group. Analyzing the group of UC patients according to the extent of ulcerative colitis, we found that the HIF-1α value was higher in all the test groups (E1, E2, and S) than in the C group with a statistically significant difference. In our study, we found that increased serum values of HIF-1α were associated with the extent of intestinal lesions, lesions that lead to the intensification of hypoxia. As a result, our data and those in the three papers listed above are consistent.

HIF-1α linked well with the degree of UC lesions in our study, with a strong and significant correlation seen between it and the TWI score in both pancolitis and left colitis forms.

Currently, the evaluation of UC activity is performed with the help of colonoscopy associated with biopsy, still considered the gold standard [67]. However, it possesses certain drawbacks, notably the potential to cause injuries during exploration, which renders it contraindicated in severe cases of UC, or it does not assist in foreseeing the recurrence of the disease regarding patients in remission.

Starting from these findings, the studies of the last decades tried to find a suitable non-invasive measurement method to avoid complications. Thus, in clinical practice, the most frequently used are the traditional non-invasive serological indicators, such as CRP, ESR, and leukocyte count, assessing the activity of inflammation. But even they cannot reflect the inflammatory activity by themselves because in other situations when an infection or tissue necrosis occurs CRP and ESR increase [68,69].

Along with serological markers, a second type of non-invasive markers is represented by the determination of fecal markers, calprotectin and lactoferrin from fecal matter, which have proven to have high sensitivity and specificity in evaluating UC activity. However, its clinical application has been restricted due to the high cost, lengthy sample processing time, and inconvenient sample collection procedures [70].

In normal physiological conditions at the level of the gastrointestinal tract, we find a balance between the immune system and the microbiota, so that the host’s immunity reacts to pathogenic antigens or tolerates harmless antigens to maintain the optimal balance [71]. Some researchers have suggested that the systemic inflammatory response may modulate the host microbiome of UC, and this has received increased attention [23,24].

In this sense, more evidence has suggested that hematological indices are considered to be faithful biomarkers for reflecting immune and inflammatory status.

Moreover, we evaluated the values of the ratio between the mean corpuscular volume and lymphocytes (MCVL) and the cumulative inflammatory index (IIC) to determine their effectiveness in evaluating disease severity compared to other factors such as NLR, PLR, and MLR in patients with UC.

We proposed to investigate the two new hematological indices starting from the findings of our colleagues [50], according to which MCVL demonstrated significant predictive value for complications with surgical risk, including abscess, necrosis, and pseudocyst. Additionally, IIC exhibited the highest predictive value for mortality, both in the pre-COVID and peri-COVID periods. These markers were more predictive than previously established inflammatory markers, which showed a specificity of less than 50% in the peri-COVID group. The authors began their analysis based on the observation that the RDW value was notably elevated in the group of patients who passed away, both in the pre-COVID and peri-COVID periods. Furthermore, the MCV values exceeded 100 in patients who succumbed in the peri-COVID group. Using these markers that summarize RBC changes that can occur in acute or chronic inflammation, they associated them with numerical changes in neutrophils, constituting components of the innate immune system, and lymphocytes, serving as markers of the adaptive immune response, and established a new prognostic marker for the severity of the acute pancreatitis patients, namely IIC [50].

The MCVL values in our study were markedly higher in the group of patients with the Mo and S UC groups and pancolitis form, while for IIC we obtained significantly higher levels only in the S UC group and pancolitis form. Moreover, we observed that the new hematological indices investigated by us correlated moderately with each other, as well as the fact that MCVL and IIC correlated strongly and very strongly with the already known inflammation indices (NLR, dNLR, SII, SIRI, AISI), in left colitis and pancolitis forms. Similarly, HIF-1α, MCVL, and IIC correlated well with the extent of UC lesions, which implies that these three biomarkers under analysis could potentially be integrated into a diagnostic algorithm and/or treatment approach, significantly influencing the progression, prognosis, and therapeutic decisions for patients with UC.

These statements are also supported by our findings related to the diagnostic accuracy of these three biomarkers. Our study evaluated whether the investigated parameters are able to discriminate between patients in the early forms of UC and those with moderate and severe forms. An optimal value, established by maximizing the sum of sensitivity and specificity for each parameter, was also determined.

Our results indicate that HIF-1α can discriminate patients with early forms at a cut-off concentration of >119.80 ng/L, with a sensitivity and specificity of 73.33% and 60.87%. With a lower specificity of 52.17%, the newly investigated hematological index, MCVL, presented an accuracy of 70.90%, suggesting that it may constitute a potential predictive factor of patients with early forms, who may develop moderate or severe forms of UC. For ICC, as well as for the other hematological indices, SIRI, SII, MLR, NLR, AISI, and dNLR, the area was <0.700.

In our study, PLR was the only one of the already-known indices that had a slightly lower diagnostic accuracy (70.04%) than MCVL. For discriminating between patients in the early forms of UC and those with moderate and severe forms, the optimal cut-off of 129.40 for PLR had a sensitivity of 73.33% and a specificity of 54.35%. The specificity obtained in our study is lower compared to those mentioned in other studies by Jeong et al. [30] (cut-off of 179.8 for PLR had a sensitivity of 35.4% and a specificity of 90.6%), or Feng et al. [58] (cut-off value was 147.96, with sensitivity and specificity of 58.3 and 75%, respectively). Also, the study of Fidan et al. [72] mentioned a cut-off value for PLR to discriminate active UC of ≥133.87, with a sensitivity of 63%, a specificity of 68%, and an AUC of 0.700. As in other studies, our results related to PLR demonstrate that platelets and lymphocytes can be a good indicator of inflammation in UC, in our case in early forms of the disease.

We acknowledge the limitations of this study, which was conducted exclusively in our reference center. Its descriptive nature and single-center design may restrict the generalizability of the results. Additionally, the relatively small sample size further impacts the study’s broader applicability. While our findings hint at potential correlations between serum levels of HIF-1α and MCVL and IIC values with disease activity, these observations would greatly benefit from an expanded study involving multiple centers. Moreover, a future prospective study could explore the dynamic correlation of serum biomarkers with conventional disease activity scores.

## 5. Conclusions

In summary, this study revealed elevated serum HIF-1α levels in UC patients, significantly higher than those in healthy controls. Moreover, the expression of HIF-1α was positively associated with the extent of disease activity, and the extent of UC lesions, suggesting that HIF-1α can be used as a new marker for evaluating UC disease activity and predicting mucosal healing. Our study has demonstrated that the MCVL and IIC were evaluated in patients with UC, and high MCVL and IIC may help doctors make decisions for patients with active UC. In our study, we observed that HIF-1α, MCVL, and PLR had the same sensitivity but HIF-1α had a much better specificity. Also, in addition to the PLR, HIF-1α and MCVL can be used as independent predictor factors in the discrimination of patients with early forms of UC. Nevertheless, large multicenter studies are expected to assess changes in inflammatory markers in larger groups of patients with UC.

## Figures and Tables

**Figure 1 biomedicines-11-03137-f001:**
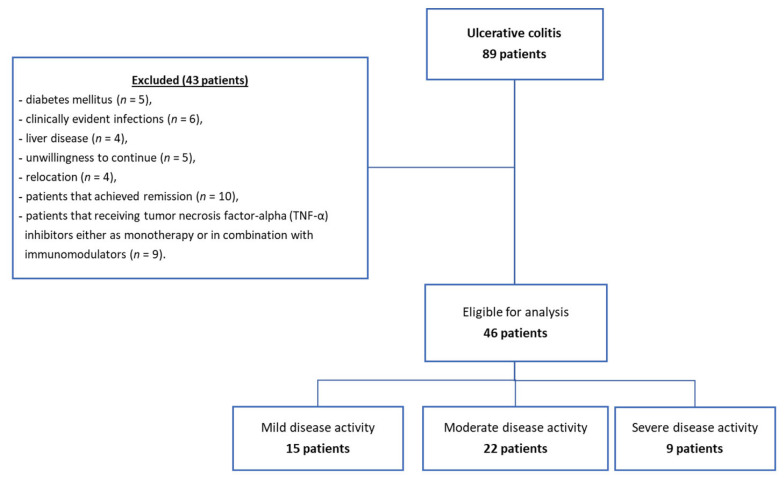
Flow chart of patient inclusion.

**Figure 2 biomedicines-11-03137-f002:**
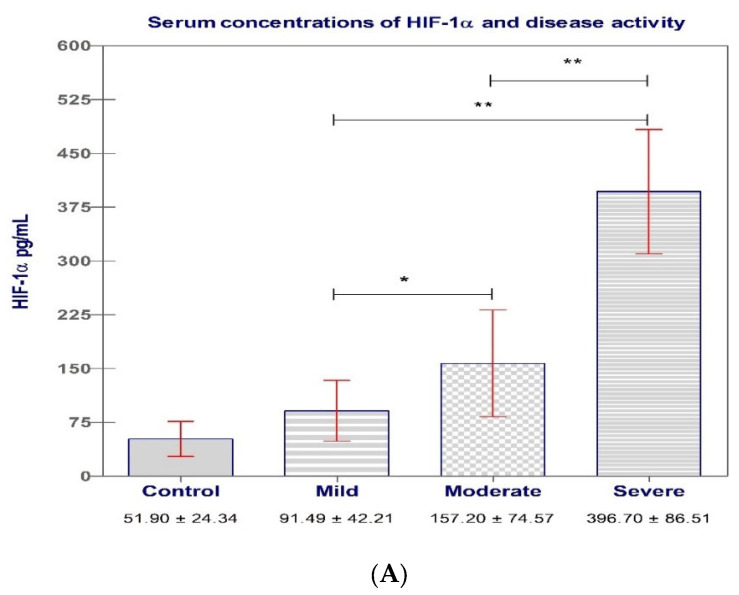
(**A**) Serum levels of HIF-1α (ng/L) and disease activity, based on TWI vs. controls. (**B**) Serum levels of HIF-1α (ng/L) and extent of disease, based on Montreal classification vs. controls; bars represent serum levels of HIF-1α from individual samples; red horizontal lines represent standard deviation. Data were analyzed for statistical significance using Kruskal—Wallis/One-way ANOVA tests between groups. *, *p* < 0.05; **, *p* < 0.0001.

**Figure 3 biomedicines-11-03137-f003:**
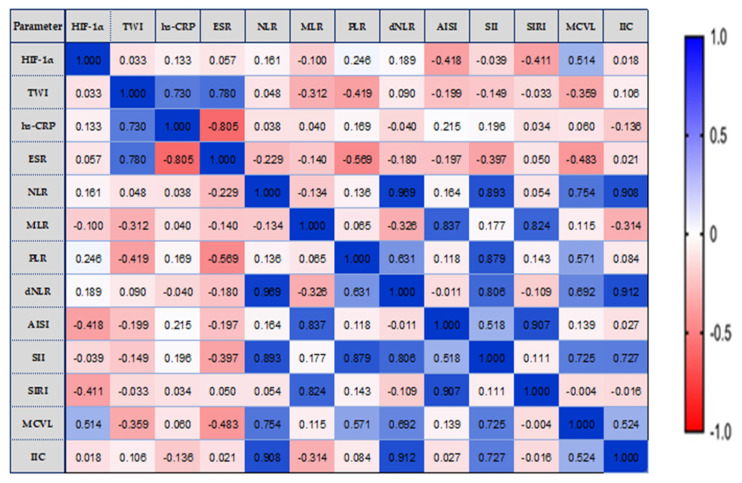
E1—The correlation heatmap illustrates the relationships between measured indicators, with colors ranging from bright blue indicating strong positive correlations to bright red indicating strong negative correlations.

**Figure 4 biomedicines-11-03137-f004:**
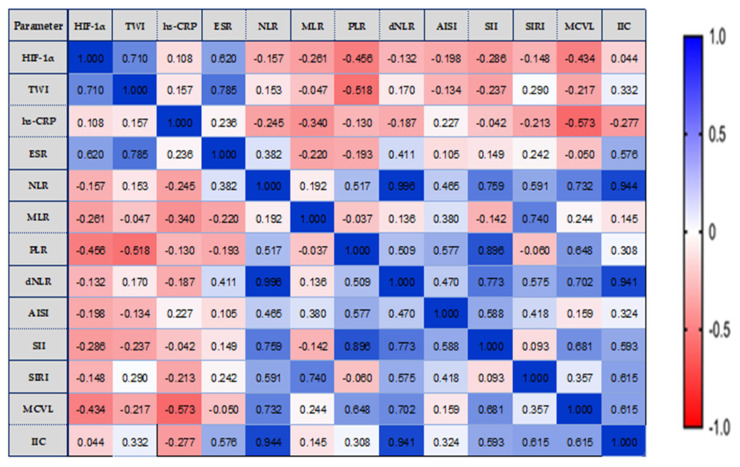
E2—The correlation heatmap illustrates the relationships between measured indicators, with colors ranging from bright blue indicating strong positive correlations to bright red indicating strong negative correlations.

**Figure 5 biomedicines-11-03137-f005:**
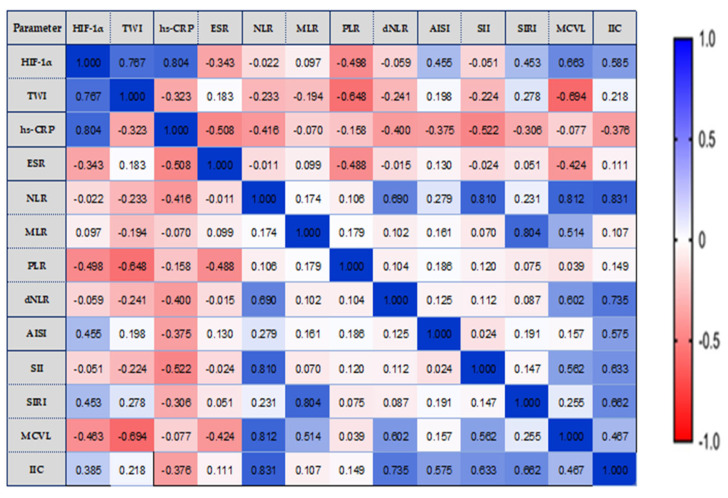
E3—The correlation heatmap illustrates the relationships between measured indicators, with colors ranging from bright blue indicating strong positive correlations to bright red indicating strong negative correlations.).

**Figure 6 biomedicines-11-03137-f006:**
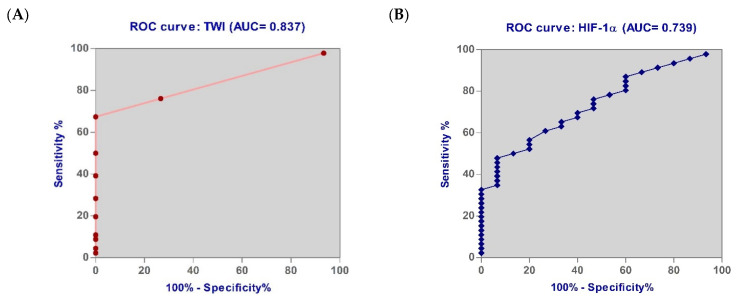
Receiver operating characteristic (ROC) curve for (**A**) TWI, (**B**) HIF-1α, (**C**) MCVL, and (**D**) PLR.

**Table 1 biomedicines-11-03137-t001:** Demographic and clinical profiles of patients included in groups UC and C.

Characteristics	UC (*N* = 46)	C (*N* = 23)
Age (year) (mean ± stdev)	49.30 ± 18.00	54.30 ± 5.75
Sex ratio (male/female) (%)	25/21 (54.35%)	13/10 (56.52%)
Residence, *n* (%)		
Urban	30 (65.22%)	14 (60.87%)
Rural	16 (34.78%)	9 (39.13%)
Montreal classification of extent of ulcerative colitis, *n* (%)		
E1—Ulcerative proctitis	15 (32.61%)	-
E2—Left-sided UC (distal UC)	18 (39.13%)	-
E3—Extensive UC (pancolitis)	13 (28.26%)	-
Truelove and Witts index (TWI), *n* (mean ± stdev)		
TWI score ≤ 5—mild activity	15 (3.20 ± 0.56)	-
TWI score = 5–9 moderate activity	22 (7.23 ± 1.15)	-
TWI score ≥ 9 severe activity	9 (11.33 ± 1.50)	-

**Table 2 biomedicines-11-03137-t002:** Serum levels of HIF-1α in relation to the disease activity and the extent of disease.

Parameter (Mean ± Stdev)	Disease Activity (TWI Severity Index)				*p*-Value from Kruskal-Wallis/ One-Way ANOVA
	Mi (*N* = 15)	Mo (*N* = 22)	S (*N* = 9)	C (*n* = 23)	
HIF-1α (ng/L)	91.49 ± 42.21	157.20 ± 74.57	396.70 ± 86.51	51.90 ± 24.34	*p <* 0.05
	**Extent of UC (Montreal classification)**				
	**E1** **(*N* = 15)**	**E2** **(*N* = 18)**	**E3** **(*N* = 13)**	**C** **(*n* = 23)**	
	93.46 ± 36.45	211.52 ± 134.68	245.64 ± 141.46	51.90 ± 24.34	

E1: Ulcerative proctitis; E2: Left-sided UC (distal UC); E3: Extensive UC (pancolitis); Mi: mild disease activity; Mo: moderate disease activity; S: severe disease activity; TWI: the Truelove and Witts severity index.

**Table 3 biomedicines-11-03137-t003:** The Inflammatory Status Biomarkers in relation to disease activity.

Parameter (Mean ± Stdev)	Mi Group (*N* = 15)	Mo Group (*N* = 22)	S Group (*N* = 9)	*p*-Value		
				Mo vs. Mi	S vs. Mi	S vs. Mo
hs-CRP (mg/dL)	16.53 ± 5.89	29.27 ± 24.30	31.56 ± 12.44	0.0240 *	0.0084 *	0.6957
ESR (mm/1st h)	20.27 ± 7.65	51.59 ± 14.34	97.33 ± 19.14	*p* < 0.0001 *	*p* < 0.0001 *	0.0002 *
FIB (mg/dL)	354.53 ± 51.79	438.86 ± 85.30	599.00 ± 110.01	0.0339 *	0.0002 *	0.0002 *
Haemoglobin (g/dL)	13.11 ± 1.19	12.38 ± 11.71	9.30 ± 1.44	0.2075	0.0006 *	0.0419 *
WBC (×10^3^/μL)	6.03 ± 0.98	8.45 ± 0.90	12.33 ± 1.18	*p* < 0.0001 *	*p* < 0.0001 *	*p* < 0.0001 *
NEU (×10^3^/μL)	4.41 ± 0.84	5.89 ± 1.05	8.43 ± 1.55	*p* < 0.0001 *	*p* < 0.0001 *	0.0001 *
LYM (×10^3^/μL)	3.36 ± 1.09	2.19 ± 0.39	1.29 ± 0.31	0.0069 *	0.0003 *	0.0024 *
MON (×10^3^/μL)	0.33 ± 0.17	0.37 ± 0.14	0.54 ± 0.33	0.4909	0.1050	0.0484 *
PLT (×10^3^/μL)	2.15 ± 0.51	2.60 ± 1.07	3.36 ±0.78	0.3694	0.0495 *	0.7901
MCV (fL)	85.81 ± 6.67	91.58 ± 4.49	102.81 ± 3.72	0.3968	*p* < 0.0001 *	0.0001 *
RDW (%)	13.40 ± 0.73	14.25 ± 1.44	18.74 ± 2.54	0.0029 *	0.0006 *	0.0041 *
NLR	2.55 ± 1.15	2.81 ± 0.86	3.58 ± 1.03	0.6641	0.0413 *	0.0517
MLR	0.16 ± 0.10	0.17 ± 0.06	0.26 ± 0.14	0.2368	0.0489 *	0.0520
PLR	66.01 ± 28.50	119.28 ± 46.50	274.84 ± 86.03	0.0124 *	0.0002 *	0.0278 *
dNLR	2.19 ± 0.92	2.40 ± 0.69	2.84 ± 0.72	0.0888	0.1062	0.0644
AISI	2364.94 ± 1027.46	2664.84 ± 1421.79	3941.33 ± 2611.54	0.1193	0.0403 *	0.0447 *
SII	560.83 ± 312.94	696.89 ± 308.38	1192.70 ± 376.13	0.0572	0.0306 *	0.0467 *
SIRI	984.93 ± 418.07	1183.45 ± 795.83	1301.41 ± 735.55	0.1033	0.0270 *	0.0656
MCVL	31.34 ± 10.98	42.91 ± 7.63	70.05 ± 16.44	0.0073 *	0.0005 *	0.0127 *
IIC	3.70 ± 1.33	4.10 ± 1.13	4.78 ± 1.83	0.6066	0.0298 *	0.0467 *

hs-CRP: high-sensitivity C-reactive protein; ESR: erythrocyte sedimentation rate; FIB: Fibrinogen; WBC: white blood cells/leukocytes; NEU: neutrophils; LYM: lymphocytes; MON: monocytes; PLT: platelets; MCV: the mean corpuscular volume; RDW: erythrocyte distribution width; NLR: neutrophil-lymphocyte ratio; MLR: monocyte-lymphocyte ratio; PLR: platelet-lymphocyte ratio; dNLR: derived neutrophil–lymphocyte ratio; AISI: aggregate index of systemic inflammation; SII: systemic immune-inflammation index; SIRI: systemic inflammatory response index; MCVL: ratio between the mean corpuscular volume/lymphocytes; IIC: cumulative inflammatory index; * *p* < 0.05: statistically significant.

**Table 4 biomedicines-11-03137-t004:** The Inflammatory Status Biomarkers in relation to the extent of disease.

Parameter (Mean ± Stdev)	E1 (*N* = 15)	E2 (*N* = 18)	E3 (*N* = 13)	*p*-Value		
				E2 vs. E1	E3 vs. E1	E3 vs. E2
hs-CRP (mg/dL)	23.17 ± 20.22	24.54 ± 22.92	29.33 ± 13.06	0.5350	0.0068 *	0.0714
ESR (mm/1st hr)	32.13 ± 19.97	50.15 ± 25.24	63.22 ± 37.05	0.0204 *	0.0120 *	0.0923
FIB (mg/dL)	371.73 ± 59.03	471.78 ± 136.93	484.31 ± 109.22	0.0391 *	0.0003 *	0.1576
Haemoglobin (g/dL)	12.82 ± 1.59	12.50 ± 0.84	11.41 ± 2.47	0.5790	0.1989	0.3366
WBC (×10^3^/μL)	7.56 ± 1.56	9.566 ± 2.98	9.69 ± 1.84	0.0176 *	0.0040 *	0.0712
NEU (×10^3^/μL)	5.572 ± 1.47	6.7 ± 2.24	6.92 ± 1.62	0.0462 *	0.0270 *	0.2650
LYM (×10^3^/μL)	2.40 ± 1.21	2.37 ± 0.82	1.70 ± 0.66	0.7160	0.0483 *	0.0418 *
MON (×10^3^/μL)	0.35 ± 0.15	0.40 ± 0.13	0.42 ± 0.27	0.5279	0.3918	0.7186
PLT (×10^3^/μL)	2.74 ± 0.79	2.89 ± 0.13	3.07 ± 0.65	0.7988	0.2218	0.8850
MCV (fL)	87.83 ± 4.55	89.62 ± 11.90	96.15 ± 8.90	0.3812	0.0466 *	0.1281
RDW (%)	14.38 ± 1.35	15.37 ± 3.03	16.21 ± 3.21	0.6906	0.1279	0.3892
NLR	3.19 ± 1.10	3.21 ± 1.44	3.32 ± 1.06	0.5017	0.0437 *	0.7177
MLR	0.17 ± 0.05	0.19 ± 0.10	0.21 ± 0.10	0.4907	0.0427 *	0.9315
PLR	139.84 ± 82.16	140.46 ± 77.05	197.48 ± 91.16	0.3398	0.0168 *	0.0264 *
dNLR	2.65 ± 1.06	2.72 ± 0.93	2.76 ± 0.89	0.8102	0.6643	1.0000
AISI	3257.40 ± 1125.00	3493.60 ± 2533.50	3500.50 ± 2023.70	0.4836	0.8538	0.5508
SII	883.65 ± 481.33	930.10 ± 472.05	1034.00 ± 434.16	0.6997	0.0267 *	0.5804
SIRI	1093.50 ± 489.63	1204.20 ± 427.51	1316.60 ± 865.57	0.0684	0.0456 *	0.0942
MCVL	42.21 ± 14.84	47.79 ± 19.80	54.96 ± 18.88	0.5089	0.0353 *	0.3028
IIC	4.20 ± 1.42	4.71 ± 2.33	4.75 ± 2.26	0.8707	0.0461 *	0.8886

hs-CRP: high-sensitivity C-reactive protein; ESR: erythrocyte sedimentation rate; FIB: Fibrinogen; WBC: white blood cells/leukocytes; NEU: neutrophils; LYM: lymphocytes; MON: monocytes; PLT: platelets; MCV: the mean corpuscular volume; RDW: erythrocyte distribution width; NLR: neutrophil-lymphocyte ratio; MLR: monocyte-lymphocyte ratio; PLR: platelet-lymphocyte ratio; dNLR: derived neutrophil–lymphocyte ratio; AISI: aggregate index of systemic inflammation; SII: systemic immune-inflammation index; SIRI: systemic inflammatory response index; MCVL: ratio between the mean corpuscular volume/lymphocytes; IIC: cumulative inflammatory index; * *p* < 0.05: statistically significant.

**Table 5 biomedicines-11-03137-t005:** Diagnostic performance of the investigated parameters.

Parameter	AUC	Cut off Values	Sensitivity %	Specificity %	*p*-Value
TWI	0.837	6.50	100.09	67.39	0.049
HIF-1α	0.739	119.80	73.33	60.87	0.036
MCVL	0.709	45.63	73.33	58.70	0.025
PLR	0.704	129.40	73.33	54.35	0.047
IIC	0.610	2.965	66.67	56.52	0.078
SIRI	0.568	791.90	60.00	63.04	0.090
SII	0.592	900.30	60.00	52.17	0.078
MLR	0.568	0.170	60.00	54.35	0.089
NLR	0.527	2.97	53.33	50.00	0.083
AISI	0.509	2783	60.00	52.17	0.091
dNLR	0.506	2.485	60.00	50.00	0.080

## Data Availability

The data used to support the findings of this study are available from the corresponding author upon reasonable request.

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
