# Peer review of "Interrelation of Hypoxia-Inducible Factor-1 Alpha (HIF-1 α) and the Ratio between the Mean Corpuscular Volume/Lymphocytes (MCVL) and the Cumulative Inflammatory Index (IIC) in Ulcerative Colitis"

_biomedicines, 2023, doi:10.3390/biomedicines11123137_

Round 1
Reviewer 1 Report
Comments and Suggestions for Authors
Existing studies have shown that HIF-1α in serum was notably higher in UC patients (73.21±28.65) than UC in remission patients (44.54±14.75) and controls (42.83±15.49. HIF-1α is likely to play an important role in the pathogenesis of UC and may serve as a biomarker to evaluate disease activity and severity in UC patients (PMID: 27073444). And author indicated that HIF-1α, MCVL and IIC can be used as a biomarker that can differentiate patients with severe or moderate disease activity. I thought that UC, as a chronic inflammatory disease, is challenging to treat effectively in later stages. Developing early-stage biomarkers is what truly matters.
Author Response
Dear Reviewer,
Thank you very much for taking the time to analyze our manuscript and for your kind appreciation and valuable suggestions.
All the typing recommended changes were performed in the body of our manuscript, with the Track Changes function activated.
Comments and Suggestions for Authors
Existing studies have shown that HIF-1α in serum was notably higher in UC patients (73.21±28.65) than UC in remission patients (44.54±14.75) and controls (42.83±15.49. HIF-1α is likely to play an important role in the pathogenesis of UC and may serve as a biomarker to evaluate disease activity and severity in UC patients (PMID: 27073444). And author indicated that HIF-1α, MCVL and IIC can be used as a biomarker that can differentiate patients with severe or moderate disease activity. I thought that UC, as a chronic inflammatory disease, is challenging to treat effectively in later stages. Developing early-stage biomarkers is what truly matters.
- We reviewed the Results (diagnostic accuracy) and Discussion section
Reviewer 2 Report
Comments and Suggestions for Authors
The paper presented by Poenariu et al addresses an important relationship between HIF1a and UC.
The writing should be improved because it is full of errors, and is not very fluid to read.
The paragraphs are too full of correlation numbers. It should be reviewed and inserted only the essential information.
The authors venture a direct relationship between HIF1a and clinical alterations, but this is not always true. The papers in the literature that support these data should be explored further, or at least better cited.
The ROC curve graphs have too basic an editing style. The legends are really too poor, and do not give an explanation of what is observed. The graphs in the other figures also present the same problems.
The results are too limited to the description of the data, with no or no critical commentary, resulting in a discussion that is too information-dense. It must be reviewed and subdivided into the Result paragraphs.
Comments on the Quality of English LanguageMany paragraph must be restructured.
Author Response
Dear Reviewer,
Thank you very much for taking the time to analyze our manuscript and for your kind appreciation and valuable suggestions.
All the typing recommended changes were performed in the body of our manuscript, with the Track Changes function activated.
Comments and Suggestions for Authors
The paper presented by Poenariu et al addresses an important relationship between HIF1a and UC.
The writing should be improved because it is full of errors, and is not very fluid to read.
- Revised
The paragraphs are too full of correlation numbers. It should be reviewed and inserted only the essential information.
- Revised
The authors venture a direct relationship between HIF1a and clinical alterations, but this is not always true. The papers in the literature that support these data should be explored further, or at least better cited.
The ROC curve graphs have too basic an editing style. The legends are really too poor, and do not give an explanation of what is observed. The graphs in the other figures also present the same problems.
- Revised
The results are too limited to the description of the data, with no or no critical commentary, resulting in a discussion that is too information-dense. It must be reviewed and subdivided into the Result paragraphs.
- Revised
Reviewer 3 Report
Comments and Suggestions for Authors
The manuscript titled “Interrelation of Hypoxia Inducible Factor-1 Alpha (HIF-1 α) and the ratio between the mean corpuscular volume / lymphocytes (MCVL) and the cumulative inflammatory index (IIC), in Ulcerative Colitis”, is very interesting and was very well written. The results were very well described. I have minor comments.
- Please improve the quality of the images in the manuscript.
- There are some sentences without references.
- This is a suggestion that does not diminish the quality of the work done. Perhaps the discussion would be more complete if the authors discussed the markers analyzed and their significance in ulcerative colitis, rather than a discussion based only on comparison with other studies.
Author Response
Dear Reviewer,
Thank you very much for taking the time to analyze our manuscript as well as for your kind appreciation and valuable suggestions.
All the typing recommended changes were performed in the body of our manuscript, with the Track Changes function activated.
Comments and Suggestions for Authors
The manuscript titled “Interrelation of Hypoxia Inducible Factor-1 Alpha (HIF-1 α) and the ratio between the mean corpuscular volume / lymphocytes (MCVL) and the cumulative inflammatory index (IIC), in Ulcerative Colitis”, is very interesting and was very well written. The results were very well described. I have minor comments.
- Please improve the quality of the images in the manuscript.
Revised
- There are some sentences without references.
Revised
- This is a suggestion that does not diminish the quality of the work done. Perhaps the discussion would be more complete if the authors discussed the markers analyzed and their significance in ulcerative colitis, rather than a discussion based only on comparison with other studies.
Revised
Round 2
Reviewer 1 Report
Comments and Suggestions for Authors
The revised version of manuscrip could be accepted.
Comments on the Quality of English Language
Have some minor errors for english language |
Author Response
Dear Reviewer,
Thank you very much for taking the time to analyze our manuscript and for your kind appreciation and valuable suggestions.
All the typing recommended changes were performed in the body of our manuscript, with the Track Changes function activated.
Have some minor errors for English language
- We have checked for errors regarding the English language

Reviewer 2 Report
Comments and Suggestions for Authors
The paper by Poenariu et al has been extensively revised.
Many linguistic errors have been corrected.
The presentation of some data has improved although still very barely sufficient.
In some cases the change made is not clear (i.e. table 5, where the values are the same).
The discussion went on too long. A lot of information should be moved to the results (as previously requested).
Finally, several literature data on the role of HIF 1a are missing (only 1 reference has been added) (as previously requested).
Comments on the Quality of English Languagesome typos error should be corrected
Author Response
Dear Reviewer,
Thank you very much for taking the time to analyze our manuscript and for your kind appreciation and valuable suggestions.
All the typing recommended changes were performed in the body of our manuscript, with the Track Changes function activated.
Comments and Suggestions for Authors
In some cases the change made is not clear (i.e. table 5, where the values are the same).
- Revised
The discussion went on too long. A lot of information should be moved to the results (as previously requested).
- Revised
Finally, several literature data on the role of HIF 1a are missing (only 1 reference has been added) (as previously requested).
